# Laundry Isolate *Delftia* sp. UBM14 Capable of Biodegrading Industrially Relevant Aminophosphonates

**DOI:** 10.3390/microorganisms12081664

**Published:** 2024-08-13

**Authors:** Ramona Riedel, Karsten Meißner, Arne Kaschubowski, Dirk Benndorf, Marion Martienssen, Burga Braun

**Affiliations:** 1Chair of Biotechnology of Water Treatment Brandenburg, Institute of Environmental Science and Environmental Technology, BTU Cottbus-Senftenberg, 03046 Cottbus, Germany; martiens@b-tu.de; 2German Environment Agency (UBA), Section II 3.3, 14195 Berlin, Germany; karsten.meissner@uba.de; 3Department of Environmental Microbiomics, University of Technology Berlin, 10587 Berlin, Germany; arne.kaschubowski@campus.tu-berlin.de (A.K.); burga.braun@tu-berlin.de (B.B.); 4Applied Biosciences and Process Engineering, Anhalt University of Applied Sciences, 06366 Köthen, Germany; dirk.benndorf@hs-anhalt.de; 5Chair of Bioprocess Engineering, Otto von Guericke University, 39106 Magdeburg, Germany; 6Bioprocess Engineering, Max Planck Institute for Dynamics of Complex Technical Systems, 39106 Magdeburg, Germany

**Keywords:** phosphonates, biodegradation, PhnX, PhnJ, PhnCDE, laundry detergents, ATMP, EDTMP, glyphosate

## Abstract

Phosphonates such as ethylenediaminetetra (methylenephosphonic acid) (EDTMP) and aminotris (methylenephosphonic acid) (ATMP) are used every day in water treatment processes or in household products. Their consumption is still increasing, regardless of the debates on their environmental impact. Here, the microbial characterisation and determination of the biodegradation potential of selected industrially relevant phosphonates for the isolate *Delftia* sp. UMB14 is reported. The opportunistic strain was isolated from a biofilm that was derived from a conventional washing machine using conventional detergents containing phosphonates. In antimicrobial susceptibility testing, the strain was only susceptible to sulfonamide, tetracycline, and chloramphenicol. Physiological and biochemical characteristics were determined using the BIOLOG EcoPlate assay. Most importantly, the strain was shown to convert D-malic acid and D-mannitol, as confirmed for strains of *Delftia lacustris,* and thus the new isolate could be closely related. Biodegradation tests with different phosphonates showed that the strain preferentially degrades ATMP and EDTMP but does not degrade glyphosate (GS) and amino (methylphosphonic acid) (AMPA). A specific gene amplification confirmed the presence of *phnX* (phosphonoacetaldehyde hydrolase) and the absence of PhnJ (the gene for the core component of C–P lyase). The presence of PhnCDE is strongly suggested for the strain, as it is common in *Delftia lacustris* species.

## 1. Introduction

Phosphonates, a large group of soluble, unreactive phosphorus compounds, have attracted much attention in recent years. There is no doubt that they have made an imperceptible contribution to our modern lifestyle. The most famous aminophosphonate, glyphosate (GS), has emerged as a global player as a systemic herbicide. However, the negative effects of GS and its transformation product amino (methylphosphonic acid) (AMPA) on human health and the environment have been the subject of intense debates for several years [1,2,3].

Independent of GS, phosphonates are used on a daily basis in many areas of modern life. This is due to their excellent complexing properties with alkaline earth and transition metals. Therefore, aminophosphonates such as ethylenediaminetetra (methylenephosphonic acid) (EDTMP) and diethylenetriamine penta (methylenephosphonic acid) (DTPMP) are now important anti-scaling and bleach-stabilising additives in detergents. Aminotris (methylenephosphonic acid) (ATMP) and hydroxyethelidene (diphosphonic acid) (HEDP) are commonly used in industrial detergents and cleaners. In addition, in the paper and textile industries, phosphonates are often used as dispersants to stabilise peroxide bleaching baths [4]. They are important components of flame-retardant compounds [5] and essential in the oilfield industry [5,6]. For the latter, complexing of barium with DTPMP is an important method to minimise or avoid the formation of incrustations on the drilling elements. Apart from technical applications, phosphonates such as HEDP and EDTMP are also successfully used in the medical treatment of bone diseases [7,8]. As a result, the use of EDTMP has become the standard treatment for bone cancer and osteoporosis [9]. The absence of phosphonates would therefore result in a dramatic loss of cleaning performance and quality, as well as medical advances in treatment.

Unfortunately, the intensive use of phosphonates in everyday life has also led to their accumulation in the aquatic environment, as they are not completely removed during wastewater treatment. This has already been reported independently by Armbruster et al. [10], Rott et al. [11,12], and Wang et al. [13]. For more than three decades, it was assumed that industrially and commercially available phosphonates such as HEDP, ATMP, EDTMP, and, especially, DTPMP were not readily biodegradable [14,15]. As recently reported, this assumption was further supported by the outcomes of the OECD test series 301 A–F for the determination of the ready biodegradability of organic compounds [16]. Synthetic phosphonates very often do not promote microbial growth due to their chemical imbalance of carbon to phosphorus. Until today, only one exception, the surfactant cocoamine bis (methylene phosphonate), meets the requirements of the test design and the conditions of the OECD test series 301 A–F. For all other phosphonates, it is not surprising that the OECD test series 301 A–F results are generally negative, as the chemical imbalance of phosphonates does not favour their biodegradation. Therefore, a new standardised biodegradation batch test has been developed to overcome the drawbacks of the OECD test series 301 A–F. Most importantly, the test can also be used to isolate novel strains from different aquatic environments, such as activated sludge and/or biofilms.

The successful application of the novel biodegradation batch test was recently reported for the novel strain *Ochrobactrum* sp. BTU1, isolated from a DTPMP standard solution [17]. The strain is able to degrade EDTMP and GS as well as common metabolites, i.e., iminodi (methylenephosphonic) (IDMP) and ethylaminobis (methylenephosphonic) acid (EABMP). Genomic and proteomic analyses confirmed that the strain degraded the synthetic phosphonates via the widespread C–P lyase pathway. The mechanism of the C–P lyase pathway was first discovered in *Escherichia coli* using the metabolic pathway from ribosyl 1,5-biphosphate to generate 5-phospho-d-ribosyl-α-1-diphosphate (PRPP) [18]. It was found that *E. coli* cannot utilise GS as a direct P source. Therefore, the C–P lyase pathway is designated as a tolerance mechanism in *E. coli* [19,20]. As recently described, the C–P lyase enzyme complex is encoded by the *phnCDEFGHIJKLMNOP* operon, within which, the PhnGHIJKLM is essential for cleavage activity. In brief, the initial catalytic step is carried out by the purine ribonucleoside triphosphate phosphonylase component PhnI, which activates the phosphonate for further catabolism. However, PhnI requires the multi-subunit protein complex (PhnGHIJK) to yield the 5′-triphosphoribosyl-1′-phosphonate [19,21]. The final C–P cleavage is then mediated by PhnJ, leading to the formation of PRPP and either sarcosine, glyoxylate, or methane for GS, AMPA, or methylphosphonate (MP), respectively.

It is now known that C–P lyase catabolism has a broad substrate specificity that allows bacteria to degrade phosphonates other than GS. Other more substrate-specific catabolisms, such as the hydrolytic pathway for the degradation of phosphonates, are also widespread in the environment and are mostly found in marine ecosystems [22]. Several hydrolases have been discovered that mediate the hydrolytic pathway and are responsible for cleaving the carbon–phosphorus (C–P) bond, such as phosphonopyruvate hydrolase (PalH) [23], phosphonoacetate hydrolase (PhnA) [24], phosphonoacetaldehyde hydrolase (PhnX) [25,26], and phosphohydrolase (PhnZ) [27,28]. According to Lockwood et al. [22], the latter three target the most abundant and ubiquitous marine biogenic phosphonate, 2-aminoethylphosphonate (2-AEP). Several bacterial strains have been reported to break down 2-AEP via PhnX [29,30,31,32]. Similar to the substrate activation in the C–P lyase pathway, the hydrolytic pathway by PhnX also requires an initial enzymatic step. The final C–P cleavage of 2-AEP by PhnX requires transamination by the 2-AEP pyruvate aminotransferase PhnW [33]. As recently stated by Zangelmi [34], the transamination reaction of PhnW is reversible, i.e., this enzyme is involved in both the biosynthesis and catabolism of phosphonates. However, PhnX does not always require PhnW as an initial enzymatic step. Recently, Zangelmi et al. [30] demonstrated that also accessory enzymes such as phosphonates breakdown factor B, C and D (PbfB, PbfB and PbfBD) can convey into the PhnWX pathway. They suggested that some of these enzymes can carry out an oxidative deamination of 2-AEP. Thus, the degradation of 2-AEP via the PhnWX pathway is not exclusively limited by PhnW, even if both PhnW and PhnX are closely associated. Further detailed information on the various enzymatic pathways, in addition to the C–P pathway, and the distinctive and unconventional characteristics of the PhnWX pathway can be found in reference [35]. Nevertheless, up to now, the biodegradation of industrially relevant phosphonates, as mentioned above, that are mediated by the PhnWX pathway has not been reported. Its narrow substrate spectrum does not seem to be predestined for phosphonates other than 2-AEP.

Since phosphonates are daily and widely used in various technical and industrial processes, especially in detergents and cleaning agents to reduce water hardness, we hypothesized that bacteria capable of degrading phosphonates can be isolated from the biofilms of washing machines. The aim of this research was to isolate and characterise a biofilm isolate for its ability to grow on industrial phosphonates that are commonly used in detergent powders, such as, ATMP, EDTMP, DTPMP, and HEDP. Additionally, GS, AMPA, and IDMP were investigated, as GS is one of the most important phosphonates, and AMPA and IDMP are major transformation products of several industrially relevant aminophosphonates. The characterisation of the strain was carried out by physiological, biochemical, and molecular analyses. The mass balance of the tested phosphonate and the minimum phosphorous requirements for bacterial growth were determined.

## 2. Materials and Methods

### 2.1. Chemicals and Reagents

HEDP, ATMP, EDTMP, and DTPMP were provided by Zschimmer & Schwarz Mohsdorf (Burgstädt, Germany). IDMP, AMPA, and GS were purchased from Sigma Aldrich (Steinheim, Germany). The chemical structures of all phosphonates are presented below (Figure 1). NH_4_COOCH_3_ was purchased from Roth (Neu-Ulm Karlsruhe, Germany). FeSO_4_ × 7H_2_O, ZnSO_4_ × 7H_2_O, H_3_BO_3_, CuCl_2_ × 2 H_2_O, NiCl_2_ × 6 H_2_O, NaMoO_4_ × 2 H_2_O, EDTA, KHCO_3_, and MgSO_4_ × 7H_2_O were purchased from Merck (Darmstadt, Germany) and KH_2_PO_4_ was purchased from Riedel-de Haën (Seelze, Germany). Nutrient agar was purchased from Roth (Karlsruhe, Germany). Nutrient broth II (NB II) was purchased from Sifin (Berlin, Germany). The local tap water (TW) of Cottbus was used for all degradation tests and provided 90 mg L^−1^ Ca^2+^, 20 mg L^−1^ Mg^2+^, 15 mg L^−1^ Na^+^, 2.5 mg L^−1^ K, 32 mg L^−1^ Cl^−^, ≤0.05 mgPO_4_^3−^ L^−1^, and 115 mg L^−1^ SO_4_^2−^. The local TW of Berlin was used for the enrichment procedure and provided 94.6 mg L^−1^ Ca^2+^, 8.75 mg L^−1^ Mg^2+^, 20 mg L^−1^ Na^+^, 5.5 mg L^−1^ K^+^, 47.5 mg L^−1^ Cl^−^, 0.04 mg TP L^−1^, and 77 mg L^−1^ SO_4_^2−^.

### 2.2. Enrichment Procedure and the Purification of Phosphonate Degrading Isolates

For the enrichment of phosphonate degrading isolates, a total of 24 biofilm samples were taken from different locations within standard washing machines. Operating of the appliance was performed with standard household laundry and commercially available detergents.

The biofilm was scraped off the detergent drawer using a sterile spatula. The removed biofilm was then suspended in a liquid medium consisting of 125 mL TW, 6 g L^−1^ ammonium acetate, 100 mg L^−1^ MgSO_4_, 125 µL SL-8 solution [36], and the investigated sole phosphorus source, with final concentrations of 50 mg L^−1^ HEDP, 100 mg L^−1^ EDTMP, or 100 mg L^−1^ DTPMP. The flasks were incubated at room temperature and shaken at 120 rpm. When the optical density (OD) at 660 nm of the culture reached >0.5, 1 mL of the culture was inoculated into fresh medium. A portion of the solution from passage 5 was applied on nutrient agar (BD DIFCO^™^, Franklin Lakes, NJ, USA). The agar plates were incubated at 25 °C, and colonies were subsequently picked and isolated from the plates. Pure cultures were obtained by passaging the cultures on these plates and verified by microscopic examinations using a light microscope (Zeiss Axioskop, Oberkochen, Germany).

### 2.3. Nucleic Acid Extraction and Sequencing of the 16S rRNA Gene

DNA was extracted from all 27 isolates using a GeneMATRIX Soil DNA Purification Kit (Roboklon, Berlin, Germany) according to the manufacturer’s instructions. PCR was performed with the personal cycler (Biometra, Göttingen, Germany). Amplification of the DNA was performed in a final volume of 50 µL, containing 25 µL 2× Color Opti Taq-PCR master mix (Roboklon, Berlin, Germany), 1 µL primer (20 pmol/µL, each), 1 µL DNA, and 22 µL PCR water. The primer pair 63f and 1387r [37] was chosen for the amplification of the 16S rRNA genes. PCR conditions were as follows: initial denaturation at 95 °C for 120 s, followed by 35 cycles, each consisting of 95 °C for 20 s, 55 °C for 30 s, and 72 °C for 120 s, terminated by a final extension step at 72 °C for 10 min. PCR amplicons were analysed under UV light after electrophoresis in 1.5% (*w*/*v*) agarose gel stained with ROTI^®^GelStain (Carl Roth, Karlsruhe, Germany). Afterwards, PCR products were purified using the GeneMATRIX Basic DNA Purification Kit (Roboklon, Berlin, Germany), following the manual. The automatic sequencing was carried out at LGC Genomics GmbH (Berlin, Germany). The quality of the obtained sequences from both primers was assessed with Chromas version 2.6.6 (https://chromas.software.informer.com/download/ accessed on 6 May 2024). Areas of particularly low quality at the beginning and end of the sequences were trimmed, exported to FASTA format, and further processed in MEGA 11 (https://www.megasoftware.net/ accessed on 6 May 2024). Sequences were aligned using MUSCLE, and the overlapping region was examined for inconsistencies. The phylogenic affiliation of the obtained 16S rDNA sequence was checked using the NCBI blastn algorithm and EzBioCloud databases.

### 2.4. Antimicrobial Susceptibility Testing

For the antibiotic susceptibility tests, the strains were grown in 8 mL of nutrient broth, followed by one passage, and then grown to the McFarland standard of 0.5. Antibiotics were chosen in accordance with [17]. Briefly, 150 µL of the culture was spread on NB agar plates. Cotton disks containing gentamycin 10 µg; ampicillin 10 µg; doxycycline 30 µg; linomycin 15 µg; sulfonamide 300 µg; polymyxin B 300 U; kanamycin 5 µg; or chloramphenicol 30 µg were placed on agar. Incubation took place at room temperature for 24 and 48 h. For evaluation of antibiotic susceptibility, the diameter of the inhibition zones was measured using calipers.

### 2.5. Physiological and Biochemical Characteristics

The BIOLOG EcoPlatesTM assay was originally created to obtain the metabolic patterns of bacterial communities. Here, BIOLOG EcoPlatesTM (Thermo Fisher Scientific (Oxoid, Hampshired, UK)) were used to evaluate the metabolic profile of a pure culture to distinguish between two closely related strains. For the BIOLOG assays with the bacterial isolate, a cell suspension was prepared as follows: an overnight suspension of the strain was washed twice in 50 mL sterile PBS, mixed, and centrifuged for 5 min at 4000× *g*. Following the cell washes, the pelleted cells were resuspended in 20 mL of sterile PBS and 150 µL aliquots corresponding to 5 × 10^8^ CFU per well (enumerated by plate counts, ref. [38] were dispensed into each of the 96 wells of the BIOLOG EcoPlate (Oxoid, Hampshire, UK)). The plates were sealed with breathe-easy sealing membrane, incubated at 28 °C, and manually scored after 8 days to determine the number of substrates utilized after 8 days. For each reading, a well was scored as positive based on a visual inspection of colour change.

### 2.6. Amplification of PhnJ and PhnX Genes

Specific primers were used to identify the C–P lyase pathway or C–P hydrolase pathways of the bacterial isolate. PCRs for each primer set were performed in a final volume of 50 µL, containing 25 µL 2× Taq-PCR master mix (Roboklon, Berlin, Germany), 2 µL primer (10 pmol/µL, each), 1 µL DNA template, and 20 µL PCR water. For the amplification of the C–P lyase key gene phnJ, the primer pairs phnJ F1/phnJ R1 and phnJ F1/phnJ R2 [39] were used with the following PCR conditions: initial denaturation at 94 °C for 120 s, followed by 35 cycles, each consisting of 94 °C for 30 s, 60 °C for 30 s, and 72 °C for 30 s, terminated by a final extension step at 72 °C for 5 min. The molecular detection of the phosphonoacetaldehyde hydrolase phnX [40] was done by using the primer pair phnX-FW and phnX-RW [41]. The PCR reaction conditions were: initial denaturation at 95 °C for 60 s, followed by 35 cycles each consisting of 95 °C for 60 s, 60 °C for 60 s, and 72 °C for 45 s, terminated by a final extension step at 72 °C for 7 min. The successful amplification of phnJ and phnX genes was analysed under UV light after electrophoresis in 1.5% (*w*/*v*) agarose gel stained with ROTI^®^GelStain (Carl Roth, Karlsruhe, Germany).

### 2.7. Biodegradation Batch Test with Delftia sp. UMB14

A single colony of pure strain *Delftia* sp. UMB14 was picked from a nutrient agar plate and incubated in 5 mL NB II medium on a rotary shaker at 180 rpm at room temperature (RT) overnight. Subsequently, 250 µL of well-suspended bacteria were transferred in 250 mL of fresh NB II medium and further incubated (180 rpm, RT, overnight). After 24 h, the bacteria were harvested by centrifuging the complete medium (5000× *g*, 10 min, RT). The bacteria were washed with sterile 0.9 % NaCl and resuspended in the degradation medium (free of phosphorus). The degradation medium was composed of sterile, filtered local tap water. The medium contained 4 g L^−1^ ammonium acetate and 2 mL L^−1^ SL-8 [35]. The SL-8 trace metal stock solution was prepared with EDTA. The OD at 660 nm of the degradation medium with washed bacteria was adjusted to an absorbance of 0.05. The liquid to headspace volume was 1:3 (v_L_/v_G_). The bacteria suspension was then continuously shaken on an orbital shaker at 180 rpm and RT (23 °C). After five days of phosphorus starvation, the bacterial suspension was diluted with a fresh degradation medium to achieve again the starting OD_660_ of 0.05. For the degradation test, the resulting phosphonate or KH_2_PO_4_ concentration was adjusted to 20 mgP L^−1^. The negative control did not contain any phosphorus source. All flasks were covered with aluminum foil before incubation at 180 rpm at RT. Samples were frequently taken for analyses of OD_660_, ortho-phosphate (o-PO_4_^3−^), total phosphorus (TP), ammonium (NH_4_^+^), total organic carbon (TOC), and total inorganic carbon (TIC). All batch tests were performed as triplicates. The development of the complete degradation test for phosphonates was described elsewhere [16].

### 2.8. Biodegradation Batch Test with Different Phosphorus Concentrations

The degradation test was prepared as described in Section 2.7, i.e., the strain was first phosphorus-starved and then diluted in fresh a degradation medium with a resulting OD_660_ of 0.05. Subsequently, either 1 mgP L^−1^, 5 mg L^−1^, 10 mgP L^−1^, 15 mg L^−1^, or 20 mg L^−1^ of phosphonate (HEDP or ATMP) or KH_2_PO_4_ as positive control was added by sterile filtration using a cellulose nitrate filter with a pore size of 0.2 µm (Sartorius, Göttingen, Germany). All flasks were incubated at 180 rpm and at RT. Samples were frequently taken for analyses of OD_660_. All batch tests were performed as triplicates.

### 2.9. Adaptation Test with ATMP or EDTMP Preconditioned Bacteria

The adaptation tests were conducted using preconditioned bacteria that had previously grown on either ATMP or EDTMP for at least 20 days. Prior to the adaptation test, the bacteria were harvested from the exponential growth phase by centrifuging the biomass (10 min, 17,000× *g*, 4 °C). The supernatant was discarded. The bacterial pellet was resuspended in fresh degradation medium without a phosphorus source and divided into eight bottles of 125 mL each. Subsequently, 20 mg P L^−1^ phosphonate was added to each bottle. The respective phosphorus sources, namely AMPA, IDMP, HEDP, GS, ATMP, EDTMP, and DTPMP, were used as the source of phosphorus. The bottles were incubated on an orbital shaker at 180 rpm and 23 °C. Samples were frequently taken for analyses of OD_660_.

### 2.10. Analytical Methods

OD_660_ was measured out using a Beckman DU 600 spectrophotometer (Fullerton, CA, USA). For TOC, o-PO_4_^3−^, TP, and NH_4_^+^ measurements, well-suspended bacterial samples were taken from flasks and centrifuged at 17,000× *g* for 10 min at 4 °C. The supernatants were collected and submitted to further analyses. Bacterial samples without centrifugation were also taken for TP analyses.

For the determination of TP, chemical digestion was carried out by adding 200 mg of “Oxisolv” (Merck, Darmstadt, Germany) into 5 mL sample volume. The sample digestion was performed by using the microwave digestion unit MARS 5 (CEM, Kamp-Lintfort, Germany). The digestion was started by linearly heating up the sample to 170 °C within 3 min and holding for another 3 min. Subsequently the samples were cooled down to RT and o-PO_4_^3−^ was measured as described above. o-PO_4_^3−^ and NH_4_^+^ were measured with a Shimadzu UV-2450 spectrophotometer (Tokyo, Japan), according to the European standard procedures DIN 38406-5:1983-10 [42] and EN ISO 6878:2004 [43]. The TOC was analysed using the TOC analyser DIMATOC-100 (Diamtech, Essen, Germany) according to DIN EN 12260 [44].

## 3. Results

### 3.1. Identification and Characterization

A total of 27 pure cultures were isolated from the phosphonate enrichment cultures. The data showed that phosphonate-degrading bacteria were obtained from the HEDP (16 isolates) and EDTMP (11 isolates) enrichments. At the family level, the isolates belonged to the families Rhiizobiaceae (9 isolates), Brucellaaceae (7 isolates), Boseaceae (1 isolate), Nitrobacteraceae (1 isolate), Paracoccaceae (4 isolates), Comamonadaceae (1 isolate), and Caulobacteraceae (4 isolates). All isolates, 27, belonged to the group of Gram (-) bacteria.

One isolate that was obtained from the HEDP enrichment was selected for further study and designated as strain UMB 14.

### 3.2. Substrate Utilization Pattern

After eight days of incubation, the formation of formazan in several wells indicated the positive metabolism of the following compounds by isolate UMB 14: L-threonine, Tween 40, D-mannitol, Tween 80, D-xylose, D-galactonic acid, D-malic acid, L-asparagine, L-phenylalanine, 4-hydroxy-benzoic acid, β-hydroxy-glycyl-L-glutamic acid, and β-keto-butyric acid. Only weak formazan formation was recorded for itaconic acid and glucose-1 phosphate.

### 3.3. Sequencing and Phylogenetic Analysis

The amplification and alignment of the 16S rDNA gene sequence of isolate UMB 14 resulted in an almost complete 16S rRNA gene, comprising 1260 nucleotides. An NCBI blast search revealed that sequence UMB 14 originated from the genus *Delftia,* and its nearest cultured neighbours are *Delftia tsuruhatensis* strain D9 (GenBank accession number MT374262.1) and *Delftia lacustris* strain NC46 (GenBank accession number MT269010.1), with a sequence identity of 100%. The 16S rRNA gene sequence of this bacterial strain was deposited in the GenBank database under accession number PP980724.

### 3.4. Antimicrobial Susceptibility Testing

For better biochemical and physiological strain characterization, we carried out antimicrobial susceptibility tests. The disk diffusion test was used to determine the susceptibility of strain *Delftia* sp. UMB 14. Antibiotics acting through the inhibition of protein synthesis, inhibition of dihydropteroate synthetase (DHPS) and alteration of membrane permeability were selected, and the inhibition zones around discs were measured. Antimicrobial susceptibility testing revealed a multidrug resistance of strain *Delftia* sp. UMB 14, with resistance to aminoglycosides (gentamycin and kanamycin), β-lactams (ampicillin), a polypeptide (polymyxin B), and lincosamide (linomycin). Susceptibility was only observed for sulphonamide (inhibition zone 2.6 ± 0.2 cm), tetracycline (doxycycline, inhibition zone 3.35 ± 0.1 cm), and chloramphenicol (inhibition zone 0.93 ± 0.2 cm).

### 3.5. Detection of phnJ and phnX Genes

Universal primers for phnJ and phnX gene amplification were used for the detection of either the C–P lyase core gene phnJ or the phosphonoacetaldehyde hydrolase gene *phnX*. Only with primer pair phnX-FW and phnX-RW, a PCR fragment of the expected amplicon length of 147 bp was obtained. Neither the primer pair phnJ F1/phnJ R1 nor phnJ F1/phnJ R2 revealed a PCR product of the expected length of 322 bp or 198 bp, respectively.

### 3.6. Biodegradation Test with Selected Synthetic Phosphonates

Biodegradation tests with seven different phosphonates as the sole P source were performed with the bacterial strain *Delftia* sp. UMB14. The strain grew differently on the provided phosphonates. In particular, no significant growth was determinable when GS and/or AMPA were the sole P source (Figure 2). The strain exhibited the most optimal growth on the phosphonates ATMP and EDTMP. On IDMP, HEDP, and DTPMP, the strain demonstrated rapid growth, particularly during the initial two days. Subsequently, the rate of growth either slowed down or kept stable, indicating no further bacterial growth. The comprehensive mass balance of the consumption of organic carbon (C_org_), nitrogen (N), and P provided a more detailed insight into the degradation test of the different phosphonates (Table 1A,B).

As above mentioned, best growth was observed for the biodegradation test of ATMP (Figure 2(A1,A2)). Also, the growth rate for ATMP was found to be the highest of all the phosphonates tested. Consequently, the shortest doubling time was determined to be 12.8 h for ATMP. The strain exhibited consumption rates of over 1300 mgC_org_ L^−1^, 120 mgN L^−1^, and nearly 4 mgP L^−1^ during the test period. In total, over 75% of the organic carbon was consumed.

The most intriguing growth occurred when EDTMP was the sole P source (Figure 2(B1,B2)). The strain exhibited growth during the initial five days, followed by a 10-day period of stagnation, which was similar to the growth pattern observed in both IDMP and GS. Following an incubation period of approximately 15 days, the strain continued to grow exponentially again. The strain achieved the second-highest final OD. The results of the mass balance also confirmed that the strain obviously utilised high amounts of C_org_ and N, i.e., 1450 mgC L^−1^ and 570 mgN L^−1^, respectively. Consequently, a total of 77% of the C_org_ was consumed through the biodegradation of EDTMP by *Delftia* sp. UMB14.

In contrast, the strain exhibited minimal growth on IDMP as a P source (Figure 2(C1,C2)). The mass balance also revealed that the strain consumed carbon, nitrogen, and phosphorus, which resulted in bacterial growth, but to a lower extent as compared with ATMP and/or EDTMP. The growth on HEDP as a P source occurred comparably with IDMP (Figure 2(D1,D2)). However, the P analysis implied rapid P reduction within the initial three days and then kept constant. In fact, the rapid P reduction was due partly to HEDP precipitation, which was visible at the bottom of the flasks. Consequently, the results of the mass balance were corrected, taking the loss of HEDP in the medium into account. An average P consumption was calculated of 2.4 mgP L^−1^. However, this result may still be incorrect. This may be further supported by the low carbon footprint compared to ATMP and EDTMP.

For DTPMP, the bacterial growth was consistently low throughout the testing period (Figure 2(E1,E2)). A slight increase in the OD was observed at the beginning and towards the end of the experiment. The data from the mass balance indicated that the strain consumed a total of approximately 250 mgC_org_ L^−1^ and 120 mgN L^−1^. Nevertheless, this represents only a low fraction of the total amount consumed by the strain in comparison to ATMP and/or EDTMP during the same period. On GS, the strain exhibited growth initially, but seemed subsequently inhibited (Figure 2(F1,F2)). After ten days of incubation, a slight growth on a very low level was observable. The results of the mass balance indicated that the strain exhibited low consumption of C_org_, N, and P. This minimal consumption indirectly corroborates the low final OD.

The results obtained for the AMPA degradation test demonstrated that indeed no phosphorus of AMPA (AMPA-P) was consumed over the measured period of 30 days (Figure 2(G1,G2)). Similar results were observed for C_org_ and N. The determined data from the mass balance thus confirmed the OD measurement. The strain was unable to grow on AMPA as a P source.

Finally, the strain was grown on KH_2_PO_4_ as a positive control (Figure 2(H1,H2)). The results clearly showed that the strain was able to grow exponentially within a few days. The bacteria consumed almost all of the C_org_, while there was still sufficient N and P available. Thus, the limited amount of available C_org_ was ultimately the limiting growth factor.

The ratio (P/S) was calculated between td of the positive control (P) and the investigated substrate (S), i.e., the individual aminophosphonates investigated. The P/S gavve a better comparison of our data and final interpretation of the determined growth rates. The results of the P/S ratio clearly indicated that the lowest growth and biodegradation in terms of C–P cleavage occurred in HEDP, DTPMP, glyphosate, and AMPA.

### 3.7. Phosphorus Requirements for Growth with Different P Sole Sources

The outcomes obtained from the previous biodegradation tests with various phosphonates clearly indicated that strain *Delftia* sp. UMB14 exhibited its best growth on ATMP as the sole P source. Nevertheless, the full analysis independently validated that only minimal amounts of P was consumed for bacterial growth. Consequently, the minimum P concentration required for unrestricted bacterial growth was determined for the aminophosphonate AMTP and compared with the organophosphonate HEDP and with the positive control KH_2_PO_4_ (Figure 3). The results demonstrate that the minimum phosphorus concentration necessary is dependent on the P-substrate provided. In the reference test, in which KH_2_PO_4_ was the only sole P source, only the lowest concentration of 1 mgP L^−1^ resulted in an early stagnation of bacterial growth due to substrate consumption (Figure 3A). The growth rates (*µ_max_*) and doubling times (*t_d_*) exhibited no significant influence with regard to the tested P concentrations for the positive control, as these were determined for the exponential growth phase (Table 2).

In contrast, a significant influence of the substrate concentration on the bacterial growth was determined for both ATMP and HEDP (Figure 3B,C). In both cases, the strain exhibited the fastest and most optimal growth at the highest concentration. Also, the final OD value was observed to be the highest for the highest substrate concentration provided. The lowest substrate concentration for both ATMP and HEDP resulted in an average doubling time of 243.5 h and 222.6 h, respectively. At the highest substrate concentration, the respective doubling times for ATMP and HEDP were determined to be 24.7 h and 35.1 h (Table 2). For both substrates, there appears to be a linear relationship between the substrate concentration and the resulting bacterial growth rate. For HEDP, the growth rate could not be determined above 15 mg L^−1^, as the phosphonate tends to precipitate in the test medium above this critical concentration.

### 3.8. Adaptation Test with ATMP and EDTMP

A third series of experiments was carried out to determine to what extent preconditioned bacteria that were previously grown on ATMP or EDTMP were able to grow on the other phosphonates used for biodegradation described in Section 3.6. Emphasis was placed on GS and AMPA, as these always resulted negatively in the previous degradation tests (Figure 4).

The results clearly show that bacterial growth on different phosphonates is possible with both preconditioning substrates, i.e., ATMP or EDTMP. In particular, the strongest bacterial growth was observed on ATMP (Figure 4A,B). However, the preconditioned bacteria also grew well on EDTMP and IDMP. Overall, the same trends were observed for the biodegradation of the phosphonates, regardless of the respective preconditioning. The EDTMP-preconditioned bacteria only seemed to show a lag phase of two days, which was not observed for the ATMP-preconditioned bacteria.

Most interestingly, no significant growth on GS and/or AMPA was detected in either test. An inhibited enzyme induction for their potential degradation is therefore ruled out, as the enzymes for phosphonate degradation were already present in the adapted bacteria due to preconditioning.

## 4. Discussion

### 4.1. Biodegradation of Phosphonates by Delftia sp. UMB14

In this study, we surveyed two different washing machines that used conventional phosphonate-based detergents. They were, therefore, suitable sources for the isolation of phosphonate-degrading bacteria. Overall, various bacteria were found in the washing machines, including in the forming biofilms. From both machines, in total, twenty-seven pure strains were isolated after specific enrichments to different phosphorous sources, such as HEDP and EDTMP. It is important to note that the capacity for phosphonate biodegradation has not been confirmed for all 27 isolates, and further investigation is required. The isolated organisms represented seven families. However, no isolates were obtained from the DTPMP enrichments. A phylogenetic analysis of isolate UMB 14 revealed 100% affiliation on an 16S rDNA gene basis to *Delftia tsuruhatensis* and *Delftia lacustris*, by NCBI Genebank searches. The latter is known as hazardous bacteria with the potential to cause ocular infection [45,46]. We cannot exclude that our isolate is also potentially capable of causing such infections. However, as we have stated above, our strain identification is currently based on the combined information of the results of the physiological and biochemical test and 16S rRNA gene sequencing. A more detailed genomic analysis is urgently required to ultimately confirm whether our isolate *Delftia* sp. UMB14 belongs to *Delftia lacustris* and has a high infection potential.

*Delftia tsuruhatensis* and *Delftia lacustris* are closely related and undistinguishable by 16S rRNA gene sequencing. For *Delftia tsuruhatensis,* antibiotic susceptibility has been observed for chloramphenicol [47], but resistance has been observed to gentamycin, ampicillin [48], and sulphonamide [47]. Also, *Delfita lacustis* was revealed to be resistant to aminoglycosides (gentamycin) [46,49]. However, Jørgensen et al. [50] were able to differentiate *Delfita lacustris* from its closest relative *Delftia tsuruhatensis* by physiological and biochemical characteristics, such as D-malic acid and D-mannitol utilisation. In accordance with Jørgensen et al. [48], strain *Delftia* sp. UMB 14 could therefore be assigned to *Delftia lacustris,* due to its utilisation of D-mannitol or D-malic acid. However, the utilisation pattern of D-malic acid and D-mannitol alike does not allow the adequate differentiation of *Deltfia lacustris* from its closest relative *Delftia tsuruhatensis*. Therefore, further genomic approaches are required to definitively identify the isolate (in progress).

In 1999, the genus *Delftia* was classified and proposed by Wen et al. [51], based on a re-evaluation of its phylogenetic and phenotypic characteristics. The two species, *Delftia tsuruhatensis* and *Delftia lacustris*, have only recently been isolated and identified, by Shigematsu et al. [52] for *Delftia tsuruhatensis* in 2003 and by Jørgensen et al. [50] for *Delftia lacustris* in 2009. Thus, only little is known on the metabolic and catabolic potential, for example, towards persistent substances. The opportunist *Delftia lacustris* has been reported to have a high potential to biodegrade organic compounds such as erythromycin [53] and pentachlorophenol [54], or to remove graffiti [55]. In addition, the potential to combat heavy metals stress through the application of metallophores [56] and the reduction of selenate and selenite has been described [57].

It is, therefore, perhaps not surprising that this opportunistic strain has been found to be able to degrade industrially relevant aminophosphonates that are commonly used in water treatments or commercial laundry detergents. The results of the present study showed that the new isolate, *Delftia* sp. UMB14, preferentially grew on ATMP and EDTMP as the sole P source. However, neither GS nor AMPA were degraded by strain *Delftia* sp. UMB14, as shown clearly by the degradation test results. Also, the result of the adaptation experiments doubtlessly confirmed that the strain could not growth on GS or AMPA as the sole P source. This observation was also supported by the results of the mass balance for C_org_, N, and P. The consumption of C_org_, for example, was a good indicator of the active metabolism of the bacteria in the batch test. The positive control showed a 97% consumption of C_org_, of 2 gC_org_ L^−1^, and 50% for P of the 20 mgP L^−1^ provided. With the supplied C, N, and P as substrates, the bacteria were able to achieve an OD_660_ value of more than 2.0 in the degradation test of the positive control. The substrate consumption determined that the strain grew unrestrictedly as long as sufficient C_org_ was available. In fact, in the case of the positive control, it was evident that C_org_ represented the limiting factor for further growth.

In comparison to the positive control, it was observed that the quantity of C_org_ consumed by the bacteria for growth on the two aminophosphonates ATMP and EDTMP was less. It is important to note, however, that significantly lower final OD values were obtained for ATMP and EDTMP, namely 1.0 and 0.5, respectively. Thus, for bacterial growth and further metabolism, more C_org_ was actually consumed when compared to the positive control. Regardless of the increased C consumption, the P turnover was reduced threefold for ATMP and fourfold for EDTMP. This indicated that the degradation of ATMP and EDTMP represented a significantly higher metabolic challenge for the strain *Delftia* sp. UMB14 compared to KH_2_PO_4_, an easily accessible P substrate.

The remaining results for IDMP, HEDP, and DTPMP similarly indicated that the strain was capable of accessing phosphonates as a phosphorus source. However, the results of the mass balance also demonstrated that a significantly lower quantity of C_org_ was consumed in comparison to ATMP and EDTMP. This evidence suggested that these structures are either more challenging for the strain to degrade or that metabolites are formed that cannot be further metabolised. As previously stated, no growth was observed for GS or AMPA. The results of the mass balance analysis unequivocally corroborated this conclusion. The fact that the strain obviously could not grow on GS or AMPA was also confirmed by the results of the genetic analysis, especially by the results of the PCR. More importantly, the determined growth rates in combination with the results of the mass balances were further evaluated and suggested the following trend from the best towards the lowest or no biodegradability of the tested phosphonates as follows: ATMP > EDTMP > IDMP > HEDP > DTPMP > GS/AMPA. This ranking contradicts the previous results of Riedel et al. [15,16] where for the pure strain *Ochrobactrum* sp. BTU1, as well as for activated sludge and TW inocula, a correlation depending on the molecule size from small to large (i.e., very good to poorly biodegradable) was determined. In these cases, however, the classical C–P lyase pathway was found to be responsible for the biodegradation of the phosphonates investigated, whose core gene *pnhJ* was not detected in strain *Delftia* sp. UMB 14. Further analyses, for example LC/MS, may help to identify potential degradation products and to better understand the degradation mechanism of the new isolate (in progress).

### 4.2. Potential Enzymatic Pathway Responsible for the Biodegrading of Phosphonates

The results showed that strain Delftia sp. UMB 14 possesses the *phnX* gene product phosphonoacetaldehyde hydrolase, which is consistent with the data from other strains of *Delftia lacustris*, e.g., *Delftia lacustris* strain FDAARGOS_890 [58]. Interestingly, the annotation data of these strains also suggested that *Delftia lacustris* possesses, on the one hand, genes associated with the phosphonate ATP-binding cassette transporter (*phnCDE*) of the C–P lyase complex and, on the other hand, does not possess the 2-AEP pyruvate aminotransferase encoded by *phnW,* as often described to be closely associated with *phnX* [32,59,60,61].

At first glance, this result seemed somewhat contradictory. However, it can also indirectly confirm the lack of degradation and growth on GS or AMPA. Both substrates are commonly known to be degradable via the C–P lyase pathway. The observation of the *phnCDE* genes of C–P lyase does not lead to confirmation of the presence of the complete C–P lyase operon. As Villarreal-Chiu et al. [60] stated, the seven gene products (PhnGHIJKLMN) constitute the core component of the C–P lyase reaction pathway. The transporter of C–P lyase, PhnCDE, with high affinity towards phosphonates, may therefore be also present for different degradation pathways, as recently demonstrated by Lee et al. [61].

Villarreal-Chiu et al. [60] evaluated in silico the distribution of the genes of the phosphonate metabolism by bacteria in the marine environment. In their analyses, they showed that nearly 50% of the sequenced genomes possessed one or more of the known phosphonate metabolic pathways. It is remarkable that four times as many strains were analysed that have genes associated to catabolic phosphonate metabolism. The C–P lyase pathway and the phosphonoacetaldehyde hydrolase pathway were both found to be most abundant among bacteria. This finding suggests that both catabolic pathways may have a very long evolutionary history. Villarreal-Chui et al. [60] showed that the distribution of the gene-encoding PhnD (periplasmic binding protein) was very broad across bacteria and even extends the distribution of the core C–P lyase genes. The authors concluded that the distribution of *phnCDE* and a number of homologues of *phnD*, especially in marine environments lacking phosphonate catabolism, suggests an extensive evolutionary rearrangement, as described by Martinez et al. [27], for the PhnY*/PhnZ pathway.

As mentioned above, the abundance of the PhnWX pathway has been suggested to be currently one of the most abundant pathways in the marine environment. However, this suggestion was based on the in-silico analyses of 1384 bacterial gene sequences and may require further confirmation. The importance of phosphonoacetaldehyde hydrolase is undoubtedly directly related to the natural product 2-AEP. As the *PhnX* operon is encoded as part of a genomic island, horizontal gene transfer is highly suggested. Villarreal-Chui et al. [60] also found an extensive rearrangement of the *phnCDE* operon components and highlighted that they are almost completely absent in pathways with a shorter evolutionary history, such as the genes of *phnA*-encoding phosphonacetate hydrolase. Thus, it may be concluded, with regard to the *Delftia lacustris,* that the presence of the genes of *phnCDE* and *phnX* are justified via horizontal gene transfer, which may also be applicable to the strain *Delftia* sp. UMB14. The absence of *phnW* for *Delftia* sp. UMB14 remains to be verified by more comprehensive genomic analyses. Currently, the draft genome analysis is in progress.

It also has to be verified how the *phnX* pathway is accessible for phosphonates different from 2-AEP, especially for industrially relevant ones, as investigated in the present study. As already stated, PhnX has a very specific substrate spectrum and requires the formation of phosphonoacetaldehyde (PnAA), which is thought to be exclusively released by PhnW and/or the three putative FAD-dependent oxidoreductases PbfB, PbfC, and PbfD [32]. The current genomic data of strain *Delftia* sp. UMB14 strongly indicate the absence of *phnW*, which is not necessarily contradictory. Zangelmi et al. showed that the absence of *phnW* is more often found in the taxonomically diverse organisms associated with *phnZ* and *phnX*, and in such cases, *pbfD* is suggested to replace *phnW* [34]. Thus, the absence of *phnW* and the presence of *phnCDE* in the genes of strains of *Delftia lacustris* may therefore be justified, but deserves more detailed genomic analyses.

### 4.3. Factors Influencing the Degradation Efficiency during Phosphonate Degradation

With regard to the biodegradation of selected industrial phosphonates by strain *Delftia* sp. UMB14, it is currently assumed that the PhnX pathway was responsible for their degradation. However, not all the phosphonates tested were degradable. The reason for this apparent specificity has not yet been identified and will be addressed by future research.

It can further be assumed that their chemical structure has an enormous influence on their biodegradation, as recently proposed by Riedel et al. [16,17]. A first indication that the chemical structure of selected phosphonates might affect their degradation efficiency in general was demonstrated for PhnD cloned from *E. coli* XL1-Blue by Rizk et al. [62]. They showed that the efficiency of different small aminophosphonates such as methylphosphonate (MP), 2-aminoethylphosphonate (2-AEP), AMPA, and GS to bind to the PhnD polypeptide varied over five orders of magnitude. Rizk et al. [62] determined the dissociation constants (Kd) for all four phosphonates, with GS having the highest value. Thus, GS had the poorest binding efficiency and the authors concluded that the compound may not be fully transported for further catabolism in the C–P lyase pathway. However, Rizk et al. [62] demonstrated this for the C–P lyase pathway, and their finding may not be fully transferable to the finding in the present study as it may be assumed that the biodegradation of ATMP and EDMPT might be even enzymatically more challenging as compared with GS.

Despite the enzymes responsible for mediating C–P cleavage, the relatively low consumption of available P due to cleavage by strain *Delftia* sp. UMB14 still raised questions and uncertainties. More specifically, the mass balance results suggested that only small amounts of P were consumed by the biodegradation of phosphonates. The highest consumption was found for ATMP, with an average of 4 mgP L^−1^. As mentioned above, the strain consumed up to 12.5 mgP L^−1^ when KH_2_PO_4_ was supplied. This discrepancy led to the general question of how much P is required for the unrestricted growth of the strain *Delftia* sp. UMB14. Interestingly, it was found that KH_2_PO_4_ and phosphonates appeared to require different threshold P concentrations. For KH_2_PO_4_, the lowest tested P concentration of 1 mgP L^−1^ resulted still in exponential growth but limited the biomass yield. In comparison, the phosphonates tested, ATMP and HEDP, showed a different relationship as they appeared to result in increased growth and biomass yield with increasing P concentration supplied. One explanation may be that both ATMP and HEDP are strong chelating agents with a high potential for interaction with alkaline earth metals. As recently shown for EDTMP, lower Ca^2+^ and Mg^2+^ exposure result in suppressed bacterial growth compared to KH_2_PO_4_ [15]. A similar phenomenon cannot be excluded for the biodegradation of ATMP and HEDP tested in this study. In addition to this finding, it has to be taken into account that the hydrolytic pathway proposed here to be responsible for the degradation of ATMP and EDTMP is highly Mg^2+^ dependent [34]. Therefore, the higher input concentration of phosphonates tested in this study may be justified. However, the reason for their higher required input concentration also remains to be verified and should be addressed in future research.

## 5. Conclusions

In this study, the inaugural description was made of the extensive diversity of bacteria (27 strains) that are capable of producing biofilms in washing machines. These biofilms have the potential to degrade detergents, including phosphonates, by colonising the surfaces of the washing machines. In addition, this is the first description of the potentially opportunistic *Delftia* sp. UMB14, isolated from a washing machine with demonstrated resistance to some antibiotics.

Sequencing and phylogenetic analyses showed that the isolate belonged to the genus *Delftia,* with 100% affiliation to *Delftia tsuruhatensis* and *Delftia lacustris* on the basis of 16S rDNA. Antimicrobial susceptibility testing showed that *Delftia lacustris* UMB14 was only susceptible to sulfonamide, tetracycline, and phenicol (chloramphenicol). The substrate utilisation pattern confirmed that strain *Delftia* sp. UMB14 utilised D-mannitol and D-malic acid, as recently reported to be characteristic of the *Delftia lacustris*. However, this utilisation pattern alone does not justify the ultimate identification of the strain. Therefore, further genomic approaches are required to definitively identify the isolate (in progress). The strain showed preferential biodegradation of the industrial aminophosphonates ATMP and EDTMP, but lacks GS and AMPA. Specific gene amplification confirmed the presence of *phnX* but not *phnJ*. Thus, the biodegradation of the studied phosphonates was not mediated by the C–P lyase pathway but is related to the phosphonoacetaldehyde hydrolase pathway. The annotation of published *Delftia lacustris* genes suggested that *Delftia* sp. UMB14 also possesses the *phnCDE* operon of the C–P lyase complex, but not the 2-AEP pyruvate aminotransferase encoded by *phnW*. However, an extensive evolutionary rearrangement by horizontal gene transfer has recently been reported and may be a reasonable explanation for the possible presence of the phnCDE and phnX genes of the isolated strain *Delftia* sp. UMB14. Similarly, the absence of phnW may initially appear contradictory, but has recently been frequently found in taxonomically diverse organisms in association with *phnZ* and *phnX*. In such cases, pbfD has been proposed to replace *phnW*, which remains to be verified for strain *Delftia* sp. UMB14.

Thus, it may be expected that the strain is also capable of degrading the industrially relevant aminophosphonates ATMP and EDTMP in the aquatic environment if the conditions for their biodegradation are similar to those in our biodegradation test.

## Figures and Tables

**Figure 1 microorganisms-12-01664-f001:**
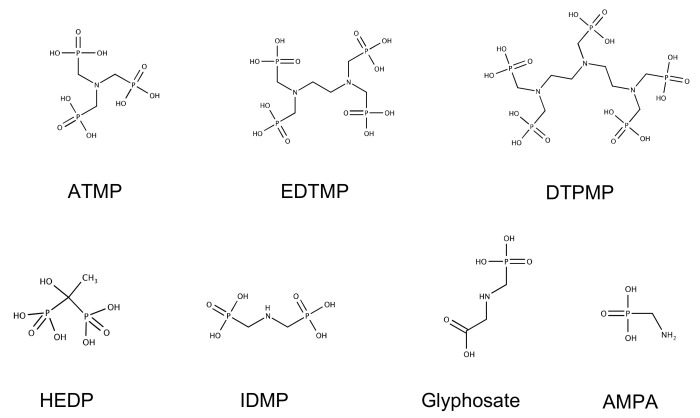
Chemical structures of the phosphonates used in the degradation tests.

**Figure 2 microorganisms-12-01664-f002:**
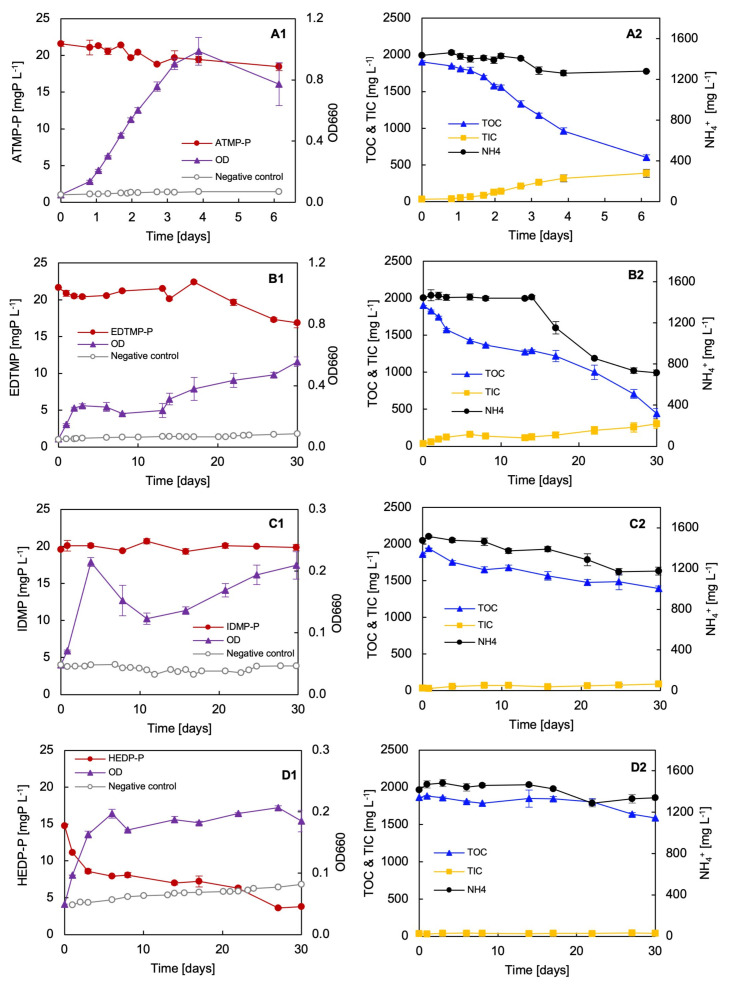
Biodegradation test of selected phosphonates with strain *Delftia* sp. UMB14. The phosphonates were dosed at 20 mgP L^−1^ to ensure comparability, except of HEDP, to prevent the precipitation of the latter. (**A1**,**A2**) ATMP. (**B1**,**B2**) EDTMP. (**C1**,**C2**) IDMP. (**D1**,**D2**) HEDP. (**E1**,**E2**) DTPMP. (**F1**,**F2**) GS. (**G1,G2**) AMPA. (**H1**,**H2**) Positive control KH_2_PO_4_. The optical density was measured at 660 nm.

**Figure 3 microorganisms-12-01664-f003:**
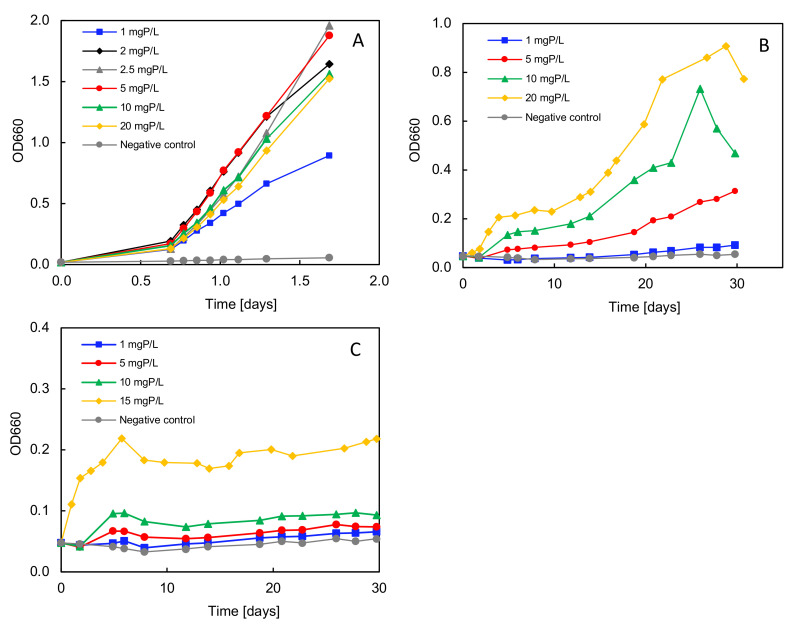
Growth test with strain *Delftia* sp. UMB14 and different sole phosphorus sources. (**A**) KH_2_PO_4_ as sole P source. (**B**) ATMP as sole P source. (**C**) HEDP as sole P source. Test was running in triplicates for each test condition. The optical density was measured at 660 nm.

**Figure 4 microorganisms-12-01664-f004:**
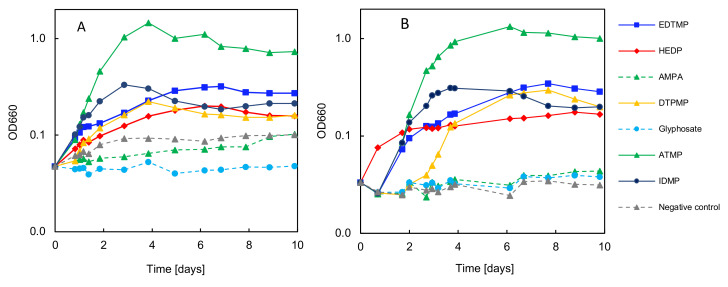
Adaptation test with inocula with preconditioned bacteria either grown on ATMP or EDTMP. (**A**) Growth test on different phosphonates with pre-inoculated bacteria on ATMP as P source. (**B**) Growth test on different phosphonates with pre-inoculated bacteria on EDTMP as P source. The optical density was measured at 660 nm.

**Table 1 microorganisms-12-01664-t001:** (**A**) Growth rates and substrate consumption by strain *Delftia* sp. UMB14 for ATMP, EDTMP, and IDMP. (**B**) Growth rates and substrate consumption by strain Delftia sp. UMB14 for HEDP, DTPMP, GS, and AMPA.

(A)					
Parameter	Unit	KH_2_PO_4_	ATMP	EDTMP	IDMP
Growth	µ [days^−1^]	3.22 ± 0.2	1.30 ± 0.1	0.80 ± 0.0	0.44 ± 0.0
	t_d_ [h]	5.2 ± 0.4	12.8 ± 0.7	20.8 ± 0.3	38.1 ± 0.6
	P/S [-]	-	0.41 ± 0.0	0.25 ± 0.0	0.14 ± 0.0
Consumption	C [mgC_org_ L^−1^]	1770.1 ± 13.8	1301.5 ± 47.8	1459.9 ± 63.4	469.1 ± 49.4
	N [mgN L^−1^]	227.0 ± 21.7	122.6 ± 9.6	567.8 ± 22.0	232.3 ± 31.3
	P [mgP L^−1^]	12.5 ± 2.1	3.9 ± 0.2	2.9 ± 0.6	1.7 ± 0.2
Efficiency	C [%]	96.6 ± 3.8	72.7 ± 2.8	76.8 ± 3.4	26.8 ± 2.9
	N [%]	18.8 ± 2.1	11.0 ± 0.8	50.2 ± 0.2	18.5 ± 2.6
	P [%]	56.5 ± 9.4	18.2 ± 0.8	13.6 ± 2.8	7.8 ± 0.9
(**B**)					
**Parameter**	**Unit**	**HEDP**	**DTPMP**	**GS**	**AMPA**
Growth	µ [days^−1^]	0.50 ± 0.0	0.18 ± 0.0	0.06 ± 0.0	-
	t_d_ [h]	33.4 ± 1.2	90.6 ± 2.6	301.5 ± 35.5	-
	P/S [-]	0.16 ± 0.0	0.06 ± 0.0	0.02 ± 0.0	-
Consumption	C [mgC_org_ L^−1^]	276.5 ± 8.9	310.3 ± 76.3	130.9 ± 18.9	32.3 ± 26.3
	N [mgN L^−1^]	58.8 ± 17.0	156.1 ± 31.0	95.1 ± 22.6	36.0 ± 9.0
	P [mgP L^−1^]	2.4 ± 0.5	0.7 ± 0.1	0.1 ± 0.3	0.2 ± 0.1
Efficiency	C [%]	15.1 ± 0.5	17.8 ± 4.4	6.8 ± 1.0	1.7 ± 1.4
	N [%]	5.8 ± 1.0	13.8 ± 2.5	7.2 ± 2.0	2.1 ± 1.0
	P [%]	16.1 ± 3.3	2.9 ± 0.3	5.0 ± 2.0	1.2 ± 0.7

**Table 2 microorganisms-12-01664-t002:** Growth rates and doubling times of *Delftia* sp. UMB14 on different P sources.

P Source	Conc. [mg L^−1^]	*µ_max_* [days^−1^]	*t_d_* [h]
KH_2_PO_4_	1	4.82 ± 0.4	3.5 ± 0.3
	2	5.15 ± 0.7	3.3 ± 0.4
	2.5	5.60 ± 1.0	3.0 ± 0.5
	5	5.50 ± 1.0	3.1 ± 0.5
	10	4.75 ± 0.1	3.5 ± 0.1
	20	5.29 ± 0.6	3.2 ± 0.3
ATMP	1	0.07 ± 0.01	243.5 ± 31.5
	5	0.22 ± 0.01	75.9 ± 4.1
	10	0.38 ± 0.00	44.1 ± 0.4
	20	0.68 ± 0.05	24.7 ± 2.0
HEDP	1	0.08 ± 0.01	222.6 ± 16.4
	5	0.16 ± 0.01	101.6 ± 1.9
	10	0.27 ± 0.00	62.7 ± 0.8
	15	0.47 ± 0.10	35.1 ± 1.3

## Data Availability

The raw data supporting the conclusions of this article will be made available by the authors on request.

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
