# Peer review of "Laundry Isolate Delftia sp. UBM14 Capable of Biodegrading Industrially Relevant Aminophosphonates"

_microorganisms, 2024, doi:10.3390/microorganisms12081664_

Round 1

Reviewer 1 Report

Comments and Suggestions for Authors

The manuscript describes the isolation, identification and characterization of a bacterial strain capable of content with different phosphonate molecules, mainly used as ingredients of detergent formulations from biofilm on washing machine. The strain denominated Delftia lacustris UMB14, was identified using the sequence of the molecular marker 16S rRNA and the observed fermentative pattern in BIOLOG EcoPlate assay. The use of potential phosphorous source of different phosphonate molecules was evaluated, as result the strains was capable of growth using methylenephosphonic acid (ATMP) and methylenephosphonic acid (HDTMP), other phosphonate ,molecules let moderate growth od strain, but glyphosate and AMPA not were employed as alternative phosphorous sources. It was proposed that Delftia lacustris UMB14 use the phosphonoacetaldehyde hydrolase (PhnX) pathway for the molecular breaking of phosphonate molecules.

Overall, the manuscript is well redacted, and the described research could be of interest to Microorganisms Journal readers. The authors need address the following comments in the manuscript.

Commentaries:

In lines 29-31, correct format of genes/operon names, must be in italics

In lines 36-38, include recent and relevant references to support the information described

In lines 65-73, “OCDE 301” could be “OECD test series 301 A-F”

In line 86, correct format of operon names, must be in italics

In line 123, review, glyphosate and AMPS are not included in detergents formulations

In lines 153 and 154, explain how the employed concentrations of the phosphate molecules were chosen

In lines 180-183, mention if this procedure was carried on for all 27 isolates

In lines 187-190, please explain, why were these antibiotics selected, what was the rationale, or is it a standard test (add reference)?

In line 201, check volume of the taken aliquots (150 mL)

In lines 202-203, close the parenthesis

In line 237, “bacteria” could be “bacterial”

In line 241, which were the sampling times?

In lines 255-265, please explain, If the phosphonates HEDP, DTMP and DTPMP were used to select the bacteria, why did they adapt with ATMP and EDTMP?

In lines 284-290, please explain, how were the genera of the isolated microorganisms determined? Was it by sequencing the 16S rRNA marker?

In lines 313-322, the results of these experiments are not presented in the manuscript, nor is it added as supplementary material, explain the relevance or applicability of the antibiotic susceptibility tests results, in the context of phosphonate degradation or bacterial strain identification.

In lines 314-315 and 318, correct format of the bacterial strain name, after first mention it could be “D. lacustris”

In lines 324-329, the results of these experiments are not presented in the manuscript, nor is it added as supplementary material.

In line 326, correct format of genes names, must be in italics

In lines 327-329, what is the length of the complete sequence of the phnX gene, what is the sequence coverage of the amplicon (147 bp) with respect to the complete size of the phnX gene, is the size of the amplified sequence sufficient to identify the phnX gene?

In figure 2, panel A1, check if “ϖ” could be “P” Y axe legend

In figure 2, all panels, “OD600” could be “OD600nm)

In figure 2 or in the methods section, please explain, how sampling times were fixed

In figure 2, in the methods section it is described that TOC, o-PO43-, TP and NH4 were determined, but in figure 2, what TIC represents?

In table 1A and 1B, what P/S means?

In figure 3, all panels, “OD600” could be “OD600nm)

In line 511, correct format of the strain name, must be in italics

In lines 515-530, please complement, after adaptation, which phosphates showed better growth, compared to previous trials?

In figure 4, all panels, “OD600” could be “OD600nm)

In line 542, it is not explained how it was determined the bacterial families of the isolates, and the result is not presented in the manuscript

In lines 546-550, is worth of mention explain how the antibiotic susceptibility profiles compare with those obtained for the Delfitia lacustris UMB14 strain, are they the same or different, how can this information be used to differentiate among Delfitia species?

In line 560, add information, is Delfitia lacustris an opportunistic pathogen?

In line 617, 649, 660-663, 721 and 725 correct format of the gene names, must be in italics

In lines 622, 624, 638, 647, 651, 664 and 724correct format of operon names, must be in italics

In line 731. complement conclusion with information about the potential of the strain in phosphonate environmental elimination, what applications are proposed and what work will be derived from this study, which are the perspectives to study?

Author Response

Response to reviewer 1

We like to thank the reviewer taking time to review our manuscript carefully. We have read your comments and agree with most of them. Please find our response below:

In lines 29-31, correct format of genes/operon names, must be in italics

Done.

In lines 36-38, include recent and relevant references to support the information described

I would like to respectfully propose that, given the already considerable number of references (approximately 60), it would be unwise to include further references here that do not contribute to improve the information in the introduction. Furthermore, the other reviewers have not identified this as a necessity, which leads us to believe that it is not an essential addition. I appreciate your understanding on this matter.

In lines 65-73, “OCDE 301” could be “OECD test series 301 A-F”

Done.

In line 86, correct format of operon names, must be in italics

Done.

In line 123, review, glyphosate and AMPS are not included in detergents formulations

We revised this sentence and added an explanation why GS, AMPA and IDMP were investigated.

In lines 153 and 154, explain how the employed concentrations of the phosphate molecules were chosen

The organophosphonate HEDP can be only applied in the degradation test up to 50 mg/L due to precipitation. This does not occur with the other phosphonates and thus higher input concentration could be chosen. We also mentioned the fact that HEDP tends to precipitate in line 361 and line 465 in the original submitted manuscript. In the M&M secition, we believe is not a good choice to give detailed explanation and justification for the chosen concentration. Since we mention the reason later in the manuscript, we prefer to avoid potential confusions and extensive explanation in section 2.2.

In lines 180-183, mention if this procedure was carried on for all 27 isolates

Yes, it was carried out with all 27 isolates. We therefore revised the first sentence of section 2.3.

In lines 187-190, please explain, why were these antibiotics selected, what was the rationale, or is it a standard test (add reference)?

The resistance test was carried out as common part of bacterial strain characterisation. We added a reference. We have chosen most antibiotics according to our recent publication investigating the biodegradation of phosphonates by strain Ochrobactrum sp. BTU1.

In line 201, check volume of the taken aliquots (150 mL)

We corrected the volume.

In lines 202-203, close the parenthesis

Done.

In line 237, “bacteria” could be “bacterial”

Done.

In line 241, which were the sampling times?

There are no fixed sampling times for any of the phosphonates analysed. The decision on the number and/or frequency of samples taken depends on the growth pattern of each phosphonate analysed. For example, sampling times of a few hours were chosen for ATMP, whereas sampling times of 24 hours or more were chosen for EDTMP. We do not consider it necessary to explain this fact in detail in the M&M section, as this information would be redundant with the results. In other words, the results show that the sampling points were individually chosen at the right time.

In lines 255-265, please explain, If the phosphonates HEDP, DTMP and DTPMP were used to select the bacteria, why did they adapt with ATMP and EDTMP?

No HEDP, DTPMP and IDMP were not used for the adaptation test. We only wanted to test whether the growth behaviour with previous adaptation on ATMP or EDTMP would be the same, especially with regard to GS and AMPA. However, detailed explanation and justification for methods this is not part of the M&M section. We therefore justified our investigations and results in section 3.8 and 4.1.

In lines 284-290, please explain, how were the genera of the isolated microorganisms determined? Was it by sequencing the 16S rRNA marker? 

This is detailed described in section 2.3 and 3.3 (The 16S rRNA gene sequence of this bacterial strain was deposited in the GenBank database under accession number PP980724).

In lines 313-322, the results of these experiments are not presented in the manuscript, nor is it added as supplementary material, explain the relevance or applicability of the antibiotic susceptibility tests results, in the context of phosphonate degradation or bacterial strain identification.

The strain shows a multidrug resistance to most antibiotics investigated. In a plot this information would not be very useful as well as in a table. Therefore, we have decided to mention the results directly in the text. Thus, the results are presented. We added an information why we have performed the tests.

In lines 314-315 and 318, correct format of the bacterial strain name, after first mention it could be “D. lacustris”

We prefer to keep the full strain name in the manuscript.

In lines 324-329, the results of these experiments are not presented in the manuscript, nor is it added as supplementary material.

This is right, but it would only show a gel with a positive amplification. Unfortunately, I cannot provide this gel image now. My co-authors who did this work are both in holydays at the moment and do not answer my Emails. We have to submit the revision until August 5, and they will have longer holydays than the submission deadline. I am very sorry and I apologies this inconvenient.

In line 326, correct format of genes names, must be in italics

Done.

In lines 327-329, what is the length of the complete sequence of the phnX gene, what is the sequence coverage of the amplicon (147 bp) with respect to the complete size of the phnX gene, is the size of the amplified sequence sufficient to identify the phnX gene?

Unfortunately, I cannot provide this information at the moment. My co-authors who did this work are both in holydays at the moment and do not answer my Emails. We have to submit the revision until August 5, and they will have longer holydays than the submission deadline. I am very sorry and I apologies this inconvenient.

In figure 2, panel A1, check if “ϖ” could be “P” Y axe legend

In figure 2, all panels, “OD600” could be “OD600nm)

We corrected panel A1 accordingly. We did not include the full wavelength in all panels to avoid misinterpretation of the OD having “nm” as unit, which would be wrong. However, we take the advice serious and added a note in the heading of Figure 2.

In figure 2 or in the methods section, please explain, how sampling times were fixed

There are no fixed sampling times for any of the phosphonates analysed. The decision on the number and/or frequency of samples taken depends on the growth pattern of each phosphonate analysed. For example, sampling times of a few hours were chosen for ATMP, whereas sampling times of 24 hours or more were chosen for EDTMP. We do not consider it necessary to explain this fact in detail in the M&M section, as this information would be redundant with the results. In other words, the results show that the sampling points were individually chosen at the right time.

In figure 2, in the methods section it is described that TOC, o-PO43-, TP and NH4 were determined, but in figure 2, what TIC represents?

The total organic carbon is measured by determining the total carbon (TC) and the total inorganic carbon (TIC). Thus, the TOC is calculated by TOC = TC – TIC. We added a note in the method section.

In table 1A and 1B, what P/S means?

The ratio (P/S) is calculated between td of the positive control (P) and the investigated substrate (S), i.e., the individual aminophosphonates investigated. The P/S give a better comparison of our data and final interpretation of determined growth rates. The results of the P/S ratio clearly indicate that the lowest growth and biodegradation in terms of C-P cleavage occurred in HEDP, DTPMP, glyphosate and AMPA. We added this information in the manuscript before Table 1.

In figure 3, all panels, “OD600” could be “OD600nm)

We did not include the full wavelength in all panels to avoid misinterpretation of the OD having “nm” as universal unit, which would be wrong. However, we take the advice serious and added a note in the heading of Figure 3.

In line 511, correct format of the strain name, must be in italics

Done.

In lines 515-530, please complement, after adaptation, which phosphates showed better growth, compared to previous trials?

This relative and currently part of additional investigations. We prefer not to induce speculation improved growth without more detailed understanding of what is the reason. In fact, most of the phosphonates resulted in increased growth rates. However, up to date there may be different theories and thesis for explanation. We prefer note to speculate about this without knowing and understanding this process in detail. The initial reason for this test was and still is to investigate whether GS and AMPA are degradable or not. The results clearly show that they are not degradable even if the enzymes are induced to degrade other aminophosphonates.

In figure 4, all panels, “OD600” could be “OD600nm)

We did not include the full wavelength in all panels to avoid misinterpretation of the OD having “nm” as universal unit, which would be wrong. However, we take the advice serious and added a note in the heading of Figure 4.

In line 542, it is not explained how it was determined the bacterial families of the isolates, and the result is not presented in the manuscript

Please section 2.3.

In lines 546-550, is worth of mention explain how the antibiotic susceptibility profiles compare with those obtained for the Delfitia lacustris UMB14 strain, are they the same or different, how can this information be used to differentiate among Delfitia species?

Unfortunately, the antibiotic susceptibility profiles cannot be used to further differentiate the strains, as several additional antibiotics would be required to make a clear classification as recently published by Eren and Güven 2022 (Biotech Studies 31(1), 36-44; http://doi.org/10.38042/biotechstudies.1103695). This would unnecessarily increase the cost of testing.

In line 560, add information, is Delfitia lacustris an opportunistic pathogen?

According to our investigation and literature review this is not a pathogen.

In line 617, 649, 660-663, 721 and 725 correct format of the gene names, must be in italics

Done.

In lines 622, 624, 638, 647, 651, 664 and 724correct format of operon names, must be in italics

Done.

In line 731. complement conclusion with information about the potential of the strain in phosphonate environmental elimination, what applications are proposed and what work will be derived from this study, which are the perspectives to study?

Any application of this strain in terms of remediation is by far to early at this stage of the investigation. We added some further information to the conclusion.

Reviewer 2 Report

Comments and Suggestions for Authors

The presented research studied possible participation of the laundry-habiting bacteria in the aminophosphonates degradation.

As the first step, authors isolated dominating bacterial strains from biofilms which were formed on the inner surface of washing machines. From the 24 biofilm samples (Line 147), there were isolated 27 strains able of the phosphonate degradation (Lines 285-289). The whole article describes characteristics and abilities of one single strain selected by authors, namely: UMB 14.

I have to point that this choice of the study object has to change emphasis of the presentation. The strain UMB 14 was identified as Delftia lacustris. This species is known as hazardous bacteria (to be more accurate – as a cause of opportunistic infections). See, for example:

Shin S.Y., Choi J.Y. & Ko K.S. Four cases of possible human infections with Delftia lacustris. Infection, 2012, 40, 709–712. https://doi.org/10.1007/s15010-012-0339-1

Sohn K.M., Baek J.Y., Cheon S., Kim Y.S. & Koo S.H. Ocular infection associated with Delftia lacustris: first report. Braz J Infect Dis, 2015, 19 (4), 449-450. doi: 10.1016/j.bjid.2015.05.001

Thus, this species is undesirable for any industrial application. On the contrary, it is necessary to look for ways to suppress these bacteria in washing machines where the clothes are treated.

In accordance with these comments, I propose that the authors 1) supplement the introduction, 2) supplement the discussion, 3) supplement the conclusion.

I ask to supplement the “Introduction” with respect to the following positions.

I. Please, give more information about diversity of bacteria that destroy phosphonates, including glyphosate, via different ways

Epiktetov D.O., Sviridov A.V., Tarlachkov S.V., Shushkova T.V., Toropygin I.Y., Leontievsky A.A. Glyphosate-induced phosphonatase operons in soil bacteria of the genus Achromobacter. Int J Mol Sci, 2024, 25 (12), 6409. doi: 10.3390/ijms25126409

etc.

Below, in results, you confirm this postulate with your own results describing the taxonomic diversity of 27 isolated strains able to degrade different phosphonates.

II. Please, mention that washing machine can accumulate various bacteria and include them in the forming biofilms. Thus, search of any possible dangerous bacteria in these biofilms was included in your study as a special goal of great significance.

I ask to supplement / expand the "Discussion" section, basing on the same positions. Please, discuss the resistance of the possible infectious bacteria Delftia lacustris according to the fact that 1) it may be a biological agent that constantly lives in the washing machine, 2) you have shown its resistance to a number of antimicrobial drugs.

At the present moment, the “Conclusion” section contains just a shortly described list of the research results. Please, help the readers and show the main points, for example: 1) it is the first description of wide diversity of bacteria (27 strains) producing biofilms in washing machines and able to live there by degrading detergents, 2) it is the first description of possible agent of opportunistic ocular infection that was isolated from the washing machine and showed its resistance to some antimicrobial drugs, 3) it is the first detailed description of the phosphonate degradation ways in Delftia lacustris.

Author Response

Response to reviewer 2

We like to thank the reviewer taking time to review our manuscript carefully. We have read your comments and agree with most of them. Please find our response below:

The presented research studied possible participation of the laundry-habiting bacteria in the aminophosphonates degradation.

As the first step, authors isolated dominating bacterial strains from biofilms which were formed on the inner surface of washing machines. From the 24 biofilm samples (Line 147), there were isolated 27 strains able of the phosphonate degradation (Lines 285-289). The whole article describes characteristics and abilities of one single strain selected by authors, namely: UMB 14. 

I have to point that this choice of the study object has to change emphasis of the presentation. The strain UMB 14 was identified as Delftia lacustris. This species is known as hazardous bacteria (to be more accurate – as a cause of opportunistic infections). See, for example: 

Shin S.Y., Choi J.Y. & Ko K.S. Four cases of possible human infections with Delftia lacustris. Infection, 2012, 40, 709–712. https://doi.org/10.1007/s15010-012-0339-1

Sohn K.M., Baek J.Y., Cheon S., Kim Y.S. & Koo S.H. Ocular infection associated with Delftia lacustris: first report. Braz J Infect Dis, 2015, 19 (4), 449-450. doi: 10.1016/j.bjid.2015.05.001 

Dear reviewer 2, we take your concern very serious and added some further information about the infection potential of this strain in the discussion including the references proposed. The second reference was already included in the original manuscript and was moved two positions forward.

Thus, this species is undesirable for any industrial application. On the contrary, it is necessary to look for ways to suppress these bacteria in washing machines where the clothes are treated.

In accordance with these comments, I propose that the authors 1) supplement the introduction, 2) supplement the discussion, 3) supplement the conclusion.

Dear reviewer 2, industrial application is not the purpose of our manuscript. This seems to be unfortunately a complete misunderstanding. The aim was to investigate the potential of bacterial strains isolated from biofilms from a conventional washing machine with regard to their potential biodegradation of phosphonates as stated in the manuscript.

I ask to supplement the “Introduction” with respect to the following positions. 

  1. Please, give more information about diversity of bacteria that destroy phosphonates, including glyphosate, via different ways 

Dear reviewer 2, up to date, there is only little literature available that provide sufficient information on the degradation pathways of the of phosphonates used in laundry. We want to keep the introduction the way it was written. In addition, the other reviewer did not give any similar recommendation. However, we followed the recommendation and added the reference and some more information on Achromobacter – the soil bacteria which is not common for washing machines.

Epiktetov D.O., Sviridov A.V., Tarlachkov S.V., Shushkova T.V., Toropygin I.Y., Leontievsky A.A. Glyphosate-induced phosphonatase operons in soil bacteria of the genus Achromobacter. Int J Mol Sci, 2024, 25 (12), 6409. doi: 10.3390/ijms25126409 

etc.

Below, in results, you confirm this postulate with your own results describing the taxonomic diversity of 27 isolated strains able to degrade different phosphonates. 

Dear reviewer 2, again this seems to be unfortunately a misunderstanding because we cannot automatically confirm that all strains are able to grow on phosphonates. However, we take your comment very serious and added a note to clarify this potential misunderstanding in the discussion.

  1. Please, mention that washing machine can accumulate various bacteria and include them in the forming biofilms.

Dear reviewer 2, we followed your recommendation and added a note in the discussion.  

Thus, search of any possible dangerous bacteria in these biofilms was included in your study as a special goal of great significance. I ask to supplement / expand the "Discussion" section, basing on the same positions. Please, discuss the resistance of the possible infectious bacteria Delftia lacustris according to the fact that 1) it may be a biological agent that constantly lives in the washing machine, 2) you have shown its resistance to a number of antimicrobial drugs. 

Dear Reviewer 2, we added a note accordingly to your and another suggestion of another reviewer which was similar to your request.

At the present moment, the “Conclusion” section contains just a shortly described list of the research results. Please, help the readers and show the main points, for example: 1) it is the first description of wide diversity of bacteria (27 strains) producing biofilms in washing machines and able to live there by degrading detergents, 2) it is the first description of possible agent of opportunistic ocular infection that was isolated from the washing machine and showed its resistance to some antimicrobial drugs, 3) it is the first detailed description of the phosphonate degradation ways in Delftia lacustris.

Dear reviewer 2, we followed your recommendation and added the missing information in the conclusion.

Reviewer 3 Report

Comments and Suggestions for Authors

The manuscript by Riedel et al. (microorganisms-3139194) is a generally well-written manuscript describing the identification of an isolate strain UMB14JZ and studying its ability to degrade phosphonates such as ATMP, EDTMP, DTPMP, HEDP, IDMP, and AMPA.

The paper may have some interest to many researchers in bioremediation field, but several issues must be clarified before it´s publication:

Lines 302-311: The 16S rRNA gene sequencing method is unable to delineate between D. tsuruhatensis and D. lacustris species. D-malic acid and D-mannitol utilization (line 552) is not adequate for to differentiate Delfita lacustris from its closest relative Delftia tsuruhatensis. Further genomic approaches are needed to clarify the species identification (next-generation sequencing, phylogenomics, comparative genomics approaches).

Using EzBioCloud database alignments, I determined that strain UMB14JZ exhibiting 100% similarity to both Delftia tsuruhatensis NBRC 16741(T) and Delftia lacustris LMG 24775(T). Only type strains should be used in the phylogenetic analyses. Please correct.

Finally, the identification of phosphonates degradation products should be interesting.

Author Response

Reviewer 3

We like to thank the reviewer taking time to review our manuscript carefully. We have read your comments and agree with most of them. Please find our response below:

The manuscript by Riedel et al. (microorganisms-3139194) is a generally well-written manuscript describing the identification of an isolate strain UMB14JZ and studying its ability to degrade phosphonates such as ATMP, EDTMP, DTPMP, HEDP, IDMP, and AMPA. 

The paper may have some interest to many researchers in bioremediation field, but several issues must be clarified before it´s publication:

Lines 302-311: The 16S rRNA gene sequencing method is unable to delineate between D. tsuruhatensis and D. lacustris species. D-malic acid and D-mannitol utilization (line 552) is not adequate for to differentiate Delfita lacustris from its closest relative Delftia tsuruhatensis. Further genomic approaches are needed to clarify the species identification (next-generation sequencing, phylogenomics, comparative genomics approaches). 

Dear reviewer 3, we agree with your statement. Currently genomic analysis is running to ultimately identify the strain. In the first attempt analysing the whole genome unfortunately failed and resulted again in 100 % coverage of Delfita lacustris and Delftia tsuruhatensis. We are running it a second time now and still the identification is a great challenge. For this reason and as stated by Jørgensen et al. 2009 a significant difference between strains is the usage of D-malic acid and D-mannitol utilisation by Delftia lacustris. To our best knowledge, we believe that combining genomic, biochemical and physiological characterisation is the best attempt for identification.

Jørgensen, N.O.; Brandt, K.K.; Nybroe, O.; Hansen, M. Delftia lacustris sp. nov., a peptidoglycan-degrading bacterium from fresh water, and emended description of Delftia tsuruhatensis as a peptidoglycan-degrading bacterium. Int. J. Syst. Evol. Microbiol. 2009, 59, 2195–2199 https//doi.org/10.1099/ijs.0.008375-0.

Using EzBioCloud database alignments, I determined that strain UMB14JZ exhibiting 100% similarity to both Delftia tsuruhatensis NBRC 16741(T) and Delftia lacustris LMG 24775(T). Only type strains should be used in the phylogenetic analyses. Please correct. 

Finally, the identification of phosphonates degradation products should be interesting.

Dear reviewer 3, The identification of degradation products of such aminophosphonates is a challenge. Up to date, only three scientific groups worldwide can perform different LC/MS approaches for such identification (Armbruster et al.; Wang et al.; Kuhn et al.). This is due to the high polarity of these compounds and several unexpected reactions during MS analysis. For the ongoing project, we will also apply the identification of potential degradation products. For the presented manuscript, the addition of further data would may be only overload the content. However, it is in progress.  

  1. Armbruster, E. Rott, R. Minke, O. Happel, Trace-level determination of phosphonates in liquid and solid phase of wastewater and environmental samples by IC-ESI-MS/MS, Anal. Bioanal. Chem. 412 (2019) 4807-4825, https://doi.org/10.1007/s00216-019-02159-5.
  2. Wang, S. Sun, C. Shan, B. Pan, Analysis of trace phosphonates in authentic water samples by pre-methylation and LC-Orbitrap MS/MS, Water Res. 161 (2019) 78-88, https://doi.org/10.1016/j.watres.2019.05.099.
  3. Kuhn, R. Jensch, I.M. Bryant, T. Fischer, S. Liebsch, M. Martienssen, Rapid sample clean-up procedure for aminophosphonate determination by LC/MS analysis, Talanta 208 (2020) 120454, 1-6, doi.org/10.1016/j.talanta. 2019.120454.
  4. Kuhn, I.M. Bryant, Martienssen M, Supplementary data on rapid sample clean-up procedure for aminophosphonate determination by LC/MS analysis, MethodsX 7 (2020) 100993, 1-11, doi.org/10.1016/j.mex.2020.100933.

Round 2

Reviewer 1 Report

Comments and Suggestions for Authors

The authors addressed the reviewers' suggestions and comments appropriately; I consider there are no additional issues to address.

Author Response

Dear reviewer 1,

On behalf of the authors, I would like to thank you very much for your afford reviewing our manuscript. Your review was very helpful for us to further improve the quality of the manuscript. Thank you very much for taking your time.

Reviewer 3 Report

Comments and Suggestions for Authors

The manuscript has not been sufficiently improved.

I had raised two basic questions that were not answered effectively.

Author Response

Dear reviewer 1,

On behalf of the authors, I would like to thank you very much for your afford reviewing our manuscript. Your review was very helpful for us to further improve the quality of the manuscript. Thank you very much for taking your time. Please find our response below:

Lines 302-311: The 16S rRNA gene sequencing method is unable to delineate between D. tsuruhatensis and D. lacustris species. D-malic acid and D-mannitol utilization (line 552) is not adequate for to differentiate Delfita lacustris from its closest relative Delftia tsuruhatensis. Further genomic approaches are needed to clarify the species identification (next-generation sequencing, phylogenomics, comparative genomics approaches). 

Dear reviewer 3, we have now carefully addressed this issue and revised both section you have mentioned. In addition, we agree that our conclusion that the isolate may be belong to strain Delftia lacustris is in fact to early without sufficient analyse the whole genome. As we stated the last time, we are currently running whole-genome sequencing a second time, but still this is in progress. You are right that only the utitlisation pattern is not sufficient and therefore we have revised the strain typing in the whole manuscript. We hope that you can not agree with this revision.

Using EzBioCloud database alignments, I determined that strain UMB14JZ exhibiting 100% similarity to both Delftia tsuruhatensis NBRC 16741(T) and Delftia lacustris LMG 24775(T). Only type strains should be used in the phylogenetic analyses. Please correct. 

Done.

Finally, the identification of phosphonates degradation products should be interesting.

As we stated that last time, we agree with your opinion. However, the LS/MS analyses is part of the on-going project and is in progress. However, we also understand that this information is missing in the manuscript and could be helpful to the reader. Therefore, we now added a note in the discussion section 4.1.
